# Criticality Controlling Mechanisms in Nematic Liquid Crystals

**DOI:** 10.3390/nano14030320

**Published:** 2024-02-05

**Authors:** Maha Zid, George Cordoyiannis, Zdravko Kutnjak, Samo Kralj

**Affiliations:** Condensed Matter Physics Department, Jožef Stefan Institute, 1000 Ljubljana, Slovenia; maha.zid@ijs.si (M.Z.); georgios.kordogiannis@ijs.si (G.C.); zdravko.kutnjak@ijs.si (Z.K.)

**Keywords:** nematic liquid crystals, critical point, critical behavior, phase transition

## Abstract

We theoretically study the generic mechanisms that could establish critical behavior in nematic liquid crystals (NLCs). The corresponding free energy density terms should exhibit linear coupling with the nematic order parameter and, via this coupling, enhance the nematic order. We consider both temperature- and pressure-driven, order–disorder phase transitions. We derive a scaled effective free energy expression that describes how qualitatively different mechanisms enforce critical behavior. Our main focus is on the impact of nanoparticles (NPs) in homogeneous NP-NLC mixtures. We illustrate that in the case of pressure-driven phase changes, lower concentrations are needed to impose critical point conditions in comparison with pure temperature variations.

## 1. Introduction

The critical points are of interest to diverse fields in physics, ranging from renormalization group theory to the electro- and thermo-mechanical response of solid and soft materials [1,2,3,4]. The proximity to a critical point strongly influences the phase transition behavior and the system’s response [5,6]. The critical behavior is determined by universal exponents, which depend on the space dimensionality and the system’s symmetry. Particular systems exhibit a critical point (CP) in their phase diagram. Typically, upon increasing an external field in a two-dimensional phase space, the first-order transition line is terminated at the CP, where the phase behavior becomes continuous, and above the CP, supercritical behavior is observed. At the CP, the system exhibits anomalous sensitivity to different stimuli [4]. Consequently, establishing CP conditions is of interest for various technological applications (e.g., sensors, caloric responses) [4,7,8,9]. 

In order to gain a profound understanding of universal features (i.e., scaling behavior) and to effectively manipulate and master the CP behavior, it is essential to identify simple experimental systems with a well-developed theoretical background. It is important to stress that normally significant external fields [4,7,8] are needed to approach CP conditions. Furthermore, applications of external fields could induce unwanted effects such as resistive Joule heating or Eddy currents if electrical or magnetic fields are used for such purposes. Sometimes, it is even difficult to impose an external field into the system.

Thermotropic uniaxial nematic liquid crystals (LCs) [10,11,12] comprise examples of experimentally approachable systems [13,14,15,16]. LCs consist of weakly interacting anisotropic molecules that exhibit several mesophases between the isotropic liquid and the crystal phases. The simplest liquid-crystalline phase is the nematic phase, characterized by a long-range uniaxial orientational order. At the mesoscopic level [10], the nematic order is described by the nematic director field $\hat{n}$ and the uniaxial nematic order parameter S. The unit vector $\hat{n}$ points along the local uniaxial order and exhibits head-to-tail invariance (i.e., the $\pm \hat{n}$ states are physically equivalent). Furthermore, *S* quantifies the degree of nematic order, which is absent for *S =* 0. In bulk equilibrium, $\hat{n}$ and *S* are spatially homogeneous and $\hat{n}$ points along a symmetry-breaking direction. In thermotropic LCs, the nematic phase is reached upon cooling from the isotropic (ordinary liquid) phase via a weakly first-order phase transition. Recently, there has been increasing interest in assessing the barocaloric effect in LCs and other soft materials [17,18,19,20], where the nematic order is obtained via a first-order phase transition by increasing pressure starting from the isotropic phase.

In nematic LCs, one can enforce critical behavior in qualitatively different ways. The most common generic triggers are the LC free energy density contributions favoring a nematic orientational order that are linearly coupled with *S*. Such a dependence exhibits external electric or magnetic field free energy terms [16] and, in common cases, LC-interfacial interactions [21,22,23,24,25] between LC molecules and LC-limiting confining substrates or immersed nanoparticles (NPs). Note that the “surface” elastic constants [10,26,27] can also exhibit such generic properties; however, for known LCs, their influence is too weak to trigger critical behavior. In the case of external fields, critical behavior is imposed for strong enough field intensities. Interfacial interactions could establish critical behavior in confined LCs, where the characteristic confinement length is comparable to the nematic order parameter correlation length [21,22,25]. Furthermore, such behavior is expected in LC-NP mixtures that, for high NP concentrations, exhibit a large enough effective NP-LC interfacial contact surface area. However, in such cases, phase separation commonly occurs [28,29], which eventually strongly reduces the effective interfacial area. In general, NPs influence the LC critical behavior via imposing variable degrees of disorder as a function of their size and concentration [14,15,30,31]. Namely, the isotropic-to-nematic (I-N) phase transition exhibits continuous symmetry breaking and such systems are extremely susceptible to disorder owing to the presence of Goldstone fluctuation modes [12,32,33]. 

Note that NP-driven critical behavior has not been observed upon varying temperature in homogeneous NP-LC mixtures, where NPs enforce a relatively weak disorder. However, a recent experimental study [20], in which orientational order–disorder phase behavior was enforced by varying pressure, reveals that a critical point can be reached in relatively diluted NP-LC mixtures. In the present paper, we theoretically study the critical behavior in NLCs using the mesoscopic Landau-de Gennes approach in terms of the tensor nematic order parameter. 

We limit our focus to cases where the nematic order is essentially uniaxial and spatially homogeneous. Both temperature- and pressure-controlled LC orders are considered. We analyze the free energy contributions that could potentially trigger critical point behavior and we derive scaled expressions predicting the onset of such conditions. Particular focus is devoted to understanding the recently observed critical point in diluted NP-LC mixtures upon varying pressure [20]. 

The structure of this paper is as follows. In Section 2, we present our mesoscopic model. In Section 3, the scaled free energy of the system is introduced describing both temperature- and pressure-driven phase behaviors. The critical behavior is analyzed in Section 4. The main findings are summarized in the conclusions. 

## 2. Model

At a mesoscopic level, one can express the nematic orientational order in terms of the traceless and symmetric tensor nematic order parameter [10,34]:(1)$$Q=\sum_{i=1}^{3}s_{i}{\hat{e}}_{i}⊗{\hat{e}}_{i}.$$
where $s_{i}$ are Q-eigenvalues (the amplitude fields) and ${\hat{e}}_{i}$ are the corresponding normalized eigenvectors. In the case of uniaxial order, where two eigenvalues are equal, Q is commonly represented as [10]
(2)$$Q^{\left(u\right)}=S\left(\hat{n}⊗\hat{n}-I/3\right)$$

The unit vector $\hat{n}$ is referred to as the nematic director (symmetry-breaking) field, *S* is the nematic order parameter amplitude field, and ***I*** stands for the unit tensor. Note that a uniaxial state can exhibit either positive or negative uniaxiality, corresponding to prolate or oblate mesoscopic order parameter geometric presentations.

We express the free energy of the system as the sum of volume and interfacial contributions:(3)$$F=\int \left(f_{c}+f_{e}+f_{f}\right)d^{3}\vec{r}+\sum_{j}\int f_{i}^{\left(j\right)}d^{2}\vec{r}$$

The first and second integral are carried out over the LC body and LC-limiting substrates, where $f_{c}$, $f_{e}$, $f_{f}$, and $f_{i}^{\left(j\right)}$ stand for the condensation, elastic, external electric or magnetic field, and *j*-th limiting interface interaction free energy densities, respectively. We express them as [10,35,36,37]
(4a)$$f_{c}=\frac{3a}{2}TrQ^{2}-\frac{9b}{2}TrQ^{3}+\frac{9c}{4}{\left(TrQ^{2}\right)}^{2}$$
(4b)$$f_{e}=L\;Tr{\left(\nabla Q\right)}^{2},$$
(4c)$$f_{f}=-\frac{3}{2}∆\chi \vec{\zeta }\times Q\vec{\zeta },$$
(4d)$$f_{i}^{\left(j\right)}=w^{\left(j\right)}\;Tr{\left(Q-Q^{\left(j\right)}\right)}^{2}.$$

Numerical coefficients in Equation (4a,c) are introduced for convenience later on. The quantities a,
*b*, and *c* are the Landau expansion coefficients; L is the representative elastic modulus (i.e., we adopt the single elastic constant approximation); $\vec{\zeta }=\zeta \hat{e}$ stands for an external electric or magnetic field aligned along a unit vector $\hat{e}$; ∆χ is the field anisotropy; and $w^{\left(j\right)}$ is the positive interfacial interaction constants of the *j*-th LC interface, which locally enforces an LC order described by the interfacial nematic tensor order parameter $Q^{\left(j\right)}$ [34,37] (i.e., in the limit $w^{\left(j\right)}\to \infty$, it holds that ${Q=Q}^{\left(j\right)}).$ In the case of an interface locally enforcing a uniaxial order, we use the following parametrization:(5)$$Q^{\left(j\right)}\left(\vec{r}\right)=S^{\left(j\right)}\left(\vec{r}\right)\left({\hat{n}}^{\left(j\right)}⊗{\hat{n}}^{\left(j\right)}-\frac{I}{3}\right).$$

Note that it is much easier to induce critical behavior by an external electric field than a magnetic field [10]. Consequently, the focus is limited to external electric field cases, where $\vec{\zeta }=\vec{E}$, $∆\chi =\varepsilon_{0}∆\varepsilon$, ∆ε stands for the dielectric anisotropy, and $\varepsilon_{0}$ is the vacuum electric permittivity. Furthermore, in modeling the interfacial interactions, we use a relatively simple interaction term weighed by a single interaction constant $w^{\left(j\right)}>0.$ In general, more complex interfacial contributions are expected. For example, the most general form of the interfacial free energy contribution up to the fourth order in Q, where one assumes that a local interface imposes a single characteristic symmetry-breaking direction ${\hat{e}}_{i},$ is given by [35]: $f_{i}^{\left(j\right)}={-w}_{1}^{\left(1\right)}\;{\hat{e}}_{i}.Q{\hat{e}}_{i}+w_{1}^{\left(2\right)}\;TrQ^{2}+w_{2}^{\left(2\right)}\;{\left({\hat{e}}_{i}.Q{\hat{e}}_{i}\right)}^{2}+w_{3}^{\left(2\right)}\;Q{\hat{e}}_{i}.Q{\hat{e}}_{i}.$ Here $w_{1}^{\left(1\right)}$, $w_{1}^{\left(2\right)},$ $w_{2}^{\left(2\right)}$, and $w_{3}^{\left(2\right)}$ stand for the bare (i.e., temperature-independent) surface interaction strengths fingerprinting a specific surface treatment procedure. In most cases, these constants are positive. This free energy density was originally proposed on symmetry grounds. Deeper insight revealing the microscopic origins of these terms are derived using a molecular mean-field theory [36]. The latter reveals that $w_{1}^{\left(1\right)}$ includes the direct substrate–LC molecule interactions and the remaining surface interaction constants emerge from the modification of interactions between LC molecules at the substrate. One reproduces Equation (4d) by enforcing $S^{\left(j\right)}=\frac{3}{2}\frac{w_{1}^{\left(1\right)}}{{3w}_{1}^{\left(2\right)}}$, ${\hat{e}}_{i}={\hat{n}}^{\left(j\right)}$ in Equation (5) [34], and setting ${\;w}_{1}^{\left(1\right)}={2S}^{\left(j\right)}w^{\left(j\right)}$, $w_{2}^{\left(2\right)}=w^{\left(j\right)},\;and\;w_{2}^{\left(2\right)}=w_{2}^{\left(3\right)}=0$, However, the essential information of interest is that in the uniaxial limit, one expects linear and quadratic contributions in terms of *S*. 

## 3. Effective Free Energy

In the following, we consider relatively dilute homogeneous LC-NP mixtures. We are interested in the impact of NPs on the I-N phase transition upon varying either the temperature or the pressure. We consider cases where NPs weakly distort the LC order and, consequently, it is reasonable to set ${Q\sim Q}^{\left(u\right)}$ (see Equation (2)). We further assume that $Q^{\left(u\right)}$ is essentially spatially homogeneous, and NP-LC interfaces play a dominant role in the LC-interfacial contributions. We assume that all NPs are equal, i.e., they impose the same anchoring conditions of the same strength ${w=w}^{\left(j\right)}$ and express the *j*-th NP contribution as
(6)$$\int f_{i}^{\left(j\right)}d^{2}\vec{r}\sim wa_{NP}\left(\frac{2}{3}\bar{S^{(j)2}}-\frac{4}{3}S\;\bar{S^{\left(j\right)}P_{2}\left(\hat{n}.{\hat{n}}^{\left(j\right)}\right)}+\frac{2}{3}S^{2}\right)=\left({w_{0}-w_{1}S+w}_{2}S^{2}\right)a_{NP}.$$
where $\bar{\left(\ldots .\right)}$ determines the average over the NP surface; $P_{2}$ is the second-order Legendre polynomial; $w_{0}=\frac{2}{3}w\bar{S^{(j)2}}$, $w_{1}=\frac{4}{3}w\;\bar{{S^{\left(j\right)}P}_{2}\left(\hat{n}\times {\hat{n}}^{\left(j\right)}\right)}$, and $w_{2}=\frac{2}{3}w$ are constants; and $a_{NP}$ stands for the nanoparticle surface area. The volume concentration of NPs is given by
(7)$$\phi =\frac{Nv_{NP}}{V},$$
where *N* stands for the number of NPs within the sample volume *V*. 

Neglecting the spatial variations in S, which are justified later, one obtains
(8)$$\frac{F}{V}\sim aS^{2}-bS^{3}+cS^{4}-\varepsilon_{0}∆\varepsilon E^{2}\langle P_{2}\left(\hat{n}.\hat{e}\right)\rangle S-\phi w_{1}\frac{a_{NP}}{v_{NP}}S+\phi w_{2}{\frac{a_{NP}}{v_{NP}}S}^{2},$$
where we limit our focus to terms depending on *S* and $\langle \ldots \rangle$ denotes averaging over the LC volume. We are interested in temperature and pressure variations across regimes where the bulk system exhibits a first-order I-N phase transition. At a constant pressure *P*, it holds that
(9)$$a\sim a_{0}\left(T-T^{*}\right),$$
where $T^{*}$ is the bulk isotropic supercooling temperature, and we neglect the temperature variations in the remaining LC material constants. 

In studying *P*-driven changes, we assume that the dominant pressure-dependent free energy contribution is also introduced by the first term in Equation (8), where we set [17]
(10)$$T^{*}\sim T_{0}^{*}+\alpha ∆P,$$
where $P=P_{0}+∆P$, $P_{0}\sim 10^{5}$ N/m^2^ stands for the atmospheric pressure, ${T^{*}(P=P_{0})=T}_{0}^{*}$, and α is constant. With this in mind, we obtain
(11)$$a\sim A_{0}\left(P^{*}-∆P\right),$$
where $P^{*}=\frac{T-T_{0}^{*}}{\alpha }$ and $A_{0}=a_{0}\alpha .$

Note that upon varying pressure, the volume of the sample changes. The *V(P)* dependence is, in general, material-dependent [38]. In our study, we limit the focus to isothermal changes for which one can reasonably assume that the ratio *P/V* is constant. Consequently, the volume concentration of NPs varies with *P* and it holds that
(12)$$\phi =\phi_{0}\frac{P}{P_{0}}=\phi_{0}\left(1+\frac{\Delta P}{P_{0}}\right),$$
where $\phi_{0}=\frac{Nv_{NP}}{V_{0}}$ and $V_{0}=VP/P_{0}$. 

### 3.1. Scaled Free Energy Density

In the following, we introduce the dimensionless free energy density $\tilde{f}$ and we use different scaling units in the temperature and pressure phase behaviors. In both cases, we use the scaled order parameter
(13)$$s=S/S_{0},\;S_{0}=\frac{b}{2c}.$$
and express the free energy in the following scaled form:(14)$$\tilde{f}=r^{\left(eff\right)}s^{2}-2s^{3}+s^{4}-\sigma^{\left(eff\right)}s.$$
where $r^{\left(eff\right)}$ stands for the effective control parameter that drives order–disorder phase transitions and $\sigma^{\left(eff\right)}$ is the effective field conjugate to the order parameter. These quantities are defined differently for temperature- and pressure-driven structural transformations. The corresponding scaling steps are described below. 

For scaling purposes, we define several material-dependent lengths. The representative NP-LC interfacial strengths are expressed in terms of the surface extrapolation lengths, defined as
(15)$$d_{e}^{\left(1\right)}=\frac{LS_{0}}{w_{1}},\;d_{e}^{\left(2\right)}=\frac{L}{w_{2}}.$$

Furthermore, we define the characteristic material-dependent lengths:(16)$$\xi_{IN}^{\left(T\right)}=\sqrt{\frac{L}{a(T_{IN}-T^{*})}},\;\xi_{IN}^{\left(P\right)}=\sqrt{\frac{L}{A_{0}(P^{*}-P_{IN})}},\;\xi_{E}=\sqrt{\frac{LS_{0}}{\varepsilon_{0}\left|∆\varepsilon \right|E^{2}}}.$$
where $\xi_{IN}^{\left(T\right)}$ and $\xi_{IN}^{\left(P\right)}$ estimate the correlation lengths in cases of temperature and pressure variations, respectively, and $\xi_{E}$ stands for the external electric field extrapolation length. We express them at the corresponding bulk phase transitions (i.e., ${T=T}_{IN}$, ${P=P}_{IN})$, where
(17)$$T_{IN}=T^{*}+\frac{b^{2}}{4a_{0}c},\;P_{IN}=P^{*}-\frac{b^{2}}{4A_{0}c}.$$

#### 3.1.1. Scaled Parameters upon Varying Temperature

In the case of temperature variations, we define the dimensionless quantities
(18a)$$r_{T}^{\left(eff\right)}=r_{T}+\phi \frac{{\xi_{IN}^{\left(T\right)}}^{2}}{d_{e}^{\left(2\right)}}\frac{a_{NP}}{v_{NP}}$$
(18b)$$r_{T}=\frac{T-T^{*}}{T_{IN}-T^{*}}$$
and
(19)$$\sigma_{T}^{\left(eff\right)}=\phi \frac{{\xi_{IN}^{\left(T\right)}}^{2}}{d_{e}^{\left(1\right)}}\frac{a_{NP}}{v_{NP}}+\frac{∆\varepsilon }{\left|∆\varepsilon \right|}\frac{{\xi_{IN}^{\left(T\right)}}^{2}}{\xi_{E}^{2}}\langle P_{2}\left(\hat{n}.\hat{e}\right)\rangle ,$$

In terms of these quantities, we obtain the expression for $\tilde{f}=f/f_{0}$, where $f_{0}=a_{0}(T_{IN}-T^{*})S_{0}^{2}$, ${r^{\left(eff\right)}=r}_{T}^{\left(eff\right)},$ and $\sigma^{\left(eff\right)}=\sigma_{T}^{\left(eff\right)}.$

#### 3.1.2. Scaled Parameters upon Varying Pressure

We further introduce the dimensionless pressure $r_{P}$ and the effective dimensionless pressure $r_{P}^{\left(eff\right)}$ as
(20a)$$r_{P}=\frac{P^{*}-∆P}{P^{*}-{∆P}_{IN}}$$
(20b)$$r_{P}^{\left(eff\right)}=r_{P}+\phi_{0}\left(1+\frac{P^{*}}{P_{0}}-\frac{∆P_{0}}{P_{0}}r_{P}\right)\frac{{\xi_{IN}^{\left(P\right)}}^{2}}{d_{e}^{\left(2\right)}}\frac{a_{NP}}{v_{NP}},$$
and the scaled surface interaction
(21)$$\sigma_{P}^{\left(eff\right)}=\phi_{0}\left(1+\frac{P^{*}}{P_{0}}-\frac{∆P_{0}}{P_{0}}r_{P}\right)\frac{{\xi_{IN}^{\left(P\right)}}^{2}}{d_{e}^{\left(1\right)}}\frac{a_{NP}}{v_{NP}}+\frac{∆\varepsilon }{\left|∆\varepsilon \right|}\frac{{\xi_{IN}^{\left(P\right)}}^{2}}{\xi_{E}^{2}}\langle P_{2}\left(\hat{n}.\hat{e}\right)\rangle ,$$ where $∆P_{0}{=P}^{*}-{∆P}_{IN}$. 

Based on the above, the expression given by Equation (14) follows for $\tilde{f}$ where $f_{0}=A_{0}\left(P^{*}-{∆P}_{IN}\right)S_{0}^{2},$ ${r^{\left(eff\right)}=r}_{P}^{\left(eff\right)},$ and $\sigma^{\left(eff\right)}=\sigma_{P}^{\left(eff\right)}.$

## 4. Phase Behavior

In this subsection, we analyze the phase behavior of systems described by the effective dimensionless free energy represented by Equation (14), and the phase transition conditions can be obtained analytically. At the phase transition, the system exhibits two equally deep minima, which can be approximately expressed by the fourth-order polynomial:(22)$$\tilde{f}=A{\left(s-s_{c}^{\left(-\right)}\right)}^{2}{\left(s-s_{c}^{\left(+\right)}\right)}^{2}+B$$
where *A* and *B* are constants, and the quantities ${\{s}_{c}^{\left(-\right)}{,s}_{c}^{\left(+\right)}\}$ determine the two minima of $\tilde{f}=\tilde{f}\left(s\right).$ The phase transition conditions are obtained by equating the expressions given by Equations (14) and (22). First-order transitions are realized when the conditions
(23a)$${r_{c}^{\left(eff\right)}=1+\sigma }^{\left(eff\right)}$$
(23b)$$\sigma^{\left(eff\right)}<\sigma_{cp}^{\left(eff\right)}.$$
are fulfilled and it holds that
(24)$$s_{c}^{\left(-\right)}=\frac{1-\sqrt{1-2\sigma^{\left(eff\right)}}}{2},\;s_{c}^{\left(+\right)}=\frac{1+\sqrt{1-2\sigma^{\left(eff\right)}}}{2}.$$
where $r_{c}^{\left(eff\right)}$ stands for the phase transition value and $\sigma_{cp}^{\left(eff\right)}$ is the CP value of $\sigma^{\left(eff\right)}$, above which the phase behavior becomes supercritical. Namely, the systems of interest exhibit critical behavior where the conditions $\frac{\partial \tilde{f}}{\partial s}=\frac{\partial^{2}\tilde{f}}{\partial s^{2}}=\frac{\partial^{3}\tilde{f}}{\partial s^{3}}=0$ are fulfilled. The CP is determined by
(25)$$\sigma_{cp}^{\left(eff\right)}=0.5,\;r_{cp}^{\left(eff\right)}=1.5,\;s_{cp}=0.5.$$

Note that the above expressions are valid for any V = V(P) dependence. In the expression below, we illustrate the quantitative behaviors for the cases where the ratio V/P is constant. In the subcritical regime, the critical phase transition temperature $T_{c}$ and the critical phase transition pressure ${∆P}_{c}$ in the corresponding temperature/pressure variations are given by
(26)$$\frac{T_{c}-T^{*}}{T_{IN}-T^{*}}=\phi \frac{{\xi_{IN}^{\left(T\right)}}^{2}}{d_{e}^{\left(1\right)}}\frac{a_{NP}}{v_{NP}}\left(1-\frac{d_{e}^{\left(1\right)}}{d_{e}^{\left(2\right)}}\right)+\frac{∆\varepsilon }{\left|∆\varepsilon \right|}\frac{{\xi_{IN}^{\left(T\right)}}^{2}}{\xi_{E}^{2}}\langle P_{2}\left(\hat{n}.\hat{e}\right)\rangle$$
(27)$$\frac{{∆P}_{IN}-{∆P}_{c}}{P^{*}-{∆P}_{IN}}=\frac{\phi_{0}\left(1+\frac{∆P_{0}}{P_{0}}\right)\;\frac{{\xi_{IN}^{\left(P\right)}}^{2}}{d_{e}^{\left(1\right)}}\frac{a_{NP}}{v_{NP}}\left(1-\frac{d_{e}^{\left(1\right)}}{d_{e}^{\left(2\right)}}\right)+\frac{∆\varepsilon }{\left|∆\varepsilon \right|}\frac{{\xi_{IN}^{\left(T\right)}}^{2}}{\xi_{E}^{2}}\langle P_{2}\left(\hat{n}.\hat{e}\right)\rangle }{1+\phi_{0}\frac{∆P_{0}}{P_{0}}\frac{{\xi_{IN}^{\left(P\right)}}^{2}}{d_{e}^{\left(1\right)}}\frac{a_{NP}}{v_{NP}}\left(1-\frac{d_{e}^{\left(1\right)}}{d_{e}^{\left(2\right)}}\right)}.$$

Note that if an external electric field $\vec{E}=E\hat{e}$ is present, the critical point condition is favored only in NLCs exhibiting a positive dielectric anisotropy. In this case, $\vec{E}$ favors the nematic director field alignment parallel to $\hat{e}$. Consequenty, one expects $\langle P_{2}(\hat{e}.\hat{n})\rangle \sim 1$ if the boundary conditions do not impose a substantially different NLC alignment. However, the equations derived above are also valid for cases where $\hat{e}$ and $\hat{n}$ are not collinear, provided that the LC states are essentially uniaxial. For NLCs exhibiting a negative dielectric anisotropy (i.e., ∆ε<0), negative uniaxiality is favored if $\left|\vec{E}\right|>0.$ Consequently, for large enough values of *E*, one expects biaxial LC states [34], which we neglect in our analysis.

In Figure 1, we plot $\tilde{f}=\tilde{f}(s)$ for different values of $\sigma^{\left(eff\right)}\leq \sigma_{cp}^{\left(eff\right)}$ at phase transition points, where both free energy minima possess an equal depth. The curvature of the *i*-th minimum is given by $c_{i}=\frac{1}{2}\frac{\partial^{2}\tilde{f}}{\partial s^{2}}\left[s_{mi}^{\left(i\right)}\right],$ where $s_{mi}^{\left(i\right)}$ describes the equilibrium value of *s* in the corresponding minimum. It holds that
(28a)$$c_{i}=1-2\sigma^{\left(eff\right)},$$
(28b)$$\Delta {\tilde{f}}_{g}=\frac{(1-2\sigma^{\left(eff\right)}{)}^{2}}{16},$$
where $\Delta {\tilde{f}}_{g}=\tilde{f}\left(s_{ma}\right)-\tilde{f}\left(s_{mi}^{\left(i\right)}\right)$ is the energy gap separating the two minima and ${s=s}_{ma}$ determines the maximum in between. Note that the order parameter correlation length for given equilibrium conditions is defined as $\xi =\sqrt{L/\left(\frac{\partial^{2}f}{\partial S^{2}}\right)}$. In our scaling, one obtains the following at $r_{c}^{\left(eff\right)}$:(29)$$\frac{\xi_{c}}{\xi_{IN}}=\frac{1}{\sqrt{1-2\sigma^{\left(eff\right)}}},$$
where $\xi_{IN}=\xi_{IN}^{\left(T\right)}$ ($\xi_{IN}=\xi_{IN}^{\left(P\right)}$) stands for the bulk correlation lengths for temperature (pressure) variations and $\xi_{c}$ represents the correlation length in systems where $\sigma^{\left(eff\right)}\leq \sigma_{cp}^{\left(eff\right)}$. Therefore, upon approaching the CP, $\xi_{c}$ diverges. Consequently, the approximation of a constant value of s becomes increasingly justifiable upon approaching $\sigma_{cp}^{\left(eff\right)}.$

In Figure 2, we illustrate the typical $s=s\left(r^{\left(eff\right)}\right)$ dependencies (Figure 2a: temperature variations, and Figure 2b: pressure variations) in the subcritical, critical, and supercritical regimes in the absence of external fields (i.e., E = 0). The bulk reference (obtained for ϕ=0) exhibits a first-order I-N phase transition, where s = 0 for $r^{\left(eff\right)}>1$. In the regime ${0<\sigma }^{\left(eff\right)}<\sigma_{cp}^{\left(eff\right)}=0.5$, the isotropic phase is replaced by a paranematic (P) phase, exhibiting a relatively weak but finite degree of orientational order. Therefore, the systems exhibit a first-order phase transition at $r_{c}^{\left(eff\right)}$ and one observes a finite degree of orientational order in the regime $r^{\left(eff\right)}>r_{c}^{\left(eff\right)}$. Furthermore, upon increasing $\sigma^{\left(eff\right)}$, the first-order transition discontinuity at $r_{c}^{\left(eff\right)}$ decreases and, eventually, vanishes at the CP.

The bistability phase regime upon varying $\sigma^{\left(eff\right)}$ is shown in Figure 3. In the bistability region, which exists in the subcritical region (i.e., $\sigma^{\left(eff\right)}<\sigma_{cp}^{\left(eff\right)}$) between the spinodal lines (denoted by the dashed and the dash-dotted curves in Figure 3), the system possesses two minima corresponding to the paranematic (P) and nematic (N) order. At the spinodal lines, one of the two minima disappears. The two minima exhibit the same depths only at the phase transition line (full line in Figure 3). The dotted line denotes the width of the bistability regime, which decreases upon increasing $\sigma^{\left(eff\right)}$ and vanishes at $\sigma^{\left(eff\right)}=\sigma_{cp}^{\left(eff\right)}$. The corresponding values of the scaled order parameters at the spinodal lines are plotted in Figure 4. In the supercritical regime ${(\sigma }^{\left(eff\right)}>\sigma_{cp}^{\left(eff\right)}),$ the system exhibits a gradual N-P transition upon varying $r^{\left(eff\right)}.$

## 5. Conclusions

We analyze how different mechanisms could shift the NLC towards the CP. The focus is on the impact of NPs on the temperature- and pressure-induced critical behavior in the first-order I-N transition. Note that CP behavior could also be imposed via a strong enough NP-induced disorder [30,31,32,33], which we do not consider in this paper. This study contributes a universal scaled form of the effective free energy describing both temperature- and pressure-driven phase transitions, in the presence of qualitatively different mechanisms that shift the system towards the critical point in the two-dimensional scaled phase space ($r^{\left(eff\right)},\sigma^{\left(eff\right)}$). The structure of $\sigma^{\left(eff\right)}$, representing the effective field conjugate to the nematic order parameter, reveals how different phase-behavior-shifting mechanisms should be quantitatively tuned to reach the CP [16,25,29]. Furthermore, we illustrate that by varying pressure, one could establish CP conditions using minute concentrations of NPs, which was recently observed experimentally [20].

In our derivation, we neglect the spatial inhomogeneities in nematic order. This approximation is sensible close to the critical point where the nematic order parameter correlation length tends to diverge [16]. Consequently, the amplitude of the nematic order exhibits only weak spatial variations. Furthermore, one assumes that nematic domains, where each is characterized by an average nematic director field symmetry-breaking direction, are relatively large. This assumption is sensible if the isotropic (paranematic)–nematic phase transition is crossed slowly upon varying the relevant phase-transition-driving parameter (i.e., temperature or pressure). On the contrary, for fast phase changes, different parts of a system in general choose different symmetry-breaking directions due to the finite speed of information propagation. Consequently, a domain-type structure is formed [39], where the characteristic linear size of domains depends on the phase transition quench rate [40]. The characteristic linear size $\xi_{p}$ of the so-called protodomains (i.e., first domains that appear after a very fast quench) is estimated by the universal Kibble–Zurek (KZ) mechanism [40]. The latter was originally introduced in cosmology [41] to explain the coarsening dynamics of the Higgs field in the early universe. Because its only ingredients are continuous symmetry breaking and causality (i.e., finite speed of information propagation), this mechanism could also be applied to condensed matter symmetry-breaking phase transitions [40], including the paranematic (isotropic)–nematic phase transformation [42]. For second-order phase transitions, the KZ mechanism predicts $\xi_{p}\sim \xi_{0}{\left(\tau_{Q}/\tau_{0}\right)}^{v/(1+\eta )},$ where $\xi_{0}$ and $\tau_{0}$ estimate the relevant order parameter correlation length and relaxation time deep in the condensed ordered phase, respectively; v and η estimate the critical coefficients of the phase transition; and $\tau_{Q}$ estimates the characteristic time in which the phase transition takes place (e.g., for temperature quenches, it is defined by $t=\tau_{Q}\frac{T-T_{c}}{T_{c}}$, where one assumes that the time *t* and temperature are linearly dependent upon crossing the phase transition temperature $T_{c}$). This equation is sensible for typical I-N phase transitions due to their weakly first-order characteristic, where it holds that v~1/2, η~1, $\xi_{0}\sim 1$ nm, and $\tau_{0}\ll 10^{-6}$ s [10,43,44]. These estimates are also applicable for relatively fast changes, if a symmetry-breaking external electric (or magnetic) field is present upon crossing the phase transition. The electric field should be strong enough, i.e., its extrapolation length $\xi_{E}$ (see Equation (16)) should be small in comparison with $\xi_{p}.$

The presented estimates are also sensible for the cases where NPs or colloids immersed in the NLC host introduce relatively strong local distortions, but not any type of disordered glass-type behavior [45,46,47,48,49,50]. In these cases, immersed particles (i.e., NPs or colloids) could mutually interact via the nematic-director-field-mediated, long-range forces, giving rise to essentially ordered particle structures [51,52,53,54,55,56] within the NLC matrix, if the NLC fluctuations are not too strong [57,58]. 

The formulation developed in this work can serve as a guide towards the development of various applications, e.g., in the area of anomalously sensitive detectors [59,60] or caloric devices [7,8], especially those based on the barocaloric effect [7,8]. In the proximity of critical points, soft materials exhibit a highly increased susceptibility to even weak external stimuli. Furthermore, susceptibilities of certain systems are related to the piezoelectric compliance [61], which is related to the electro-mechanical response. Consequently, mastering the critical point conditions is interesting for diverse thermal (e.g., heat management) applications, actuators, and sensitive sensors [7,8]. In these effects, the adiabatic variation in an external (e.g., electric, magnetic, elastic) field enforces a large temperature change that can be maximized near critical points [4]. Therefore, this work suggests how the combination of different phase-behavior-shifting mechanisms could be exploited in such applications.

## Figures and Tables

**Figure 1 nanomaterials-14-00320-f001:**
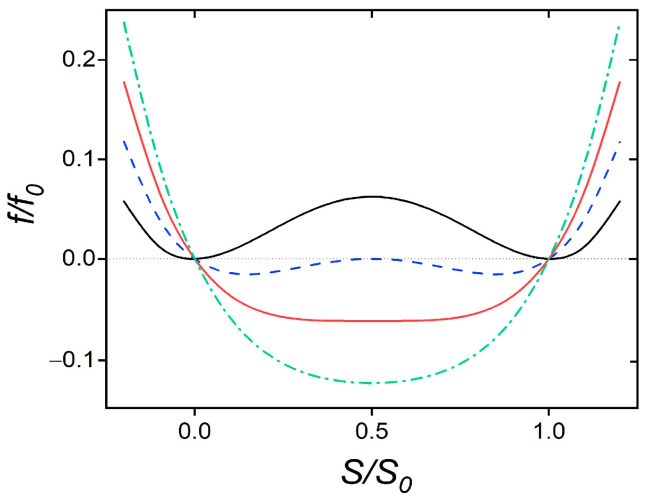
The dimensionless free energy density $f/f_{0}$ is plotted as a function of the scaled order parameter $s=S/S_{0}$. Black solid line: bulk phase transition; blue dashed line: phase transition for $\sigma^{\left(eff\right)}=0.25$; red solid line: critical point, $\sigma^{\left(eff\right)}=\sigma_{cp}^{\left(eff\right)}\equiv 0.5$; green dash-dotted line: supercritical state, $\sigma^{\left(eff\right)}=0.75.$

**Figure 2 nanomaterials-14-00320-f002:**
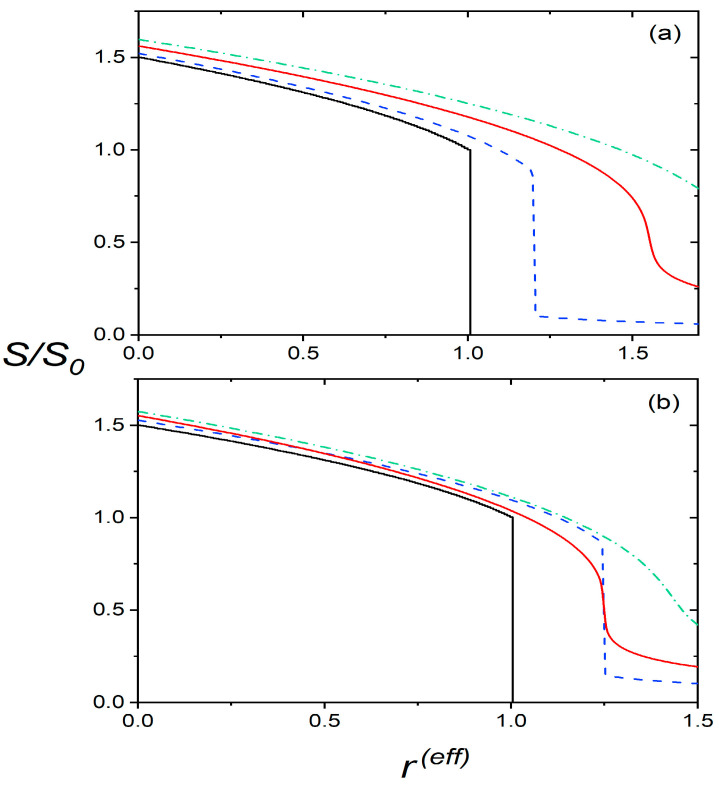
The scaled order parameter $s=S/S_{0}$ is plotted as a function of the effective reduced temperature for different effective field strengths. For illustration purposes, we refer only to the impact of NPs, and we set $w_{2}=0$ (in these cases, $r_{T}=r_{T}^{\left(eff\right)}$ and $r_{P}=r_{P}^{\left(eff\right)}$). Upper panel (**a**): temperature variation ($r^{\left(eff\right)}=r_{T}^{\left(eff\right)}$). Black solid line: $\sigma^{\left(eff\right)}=0;$ blue dashed line: $\sigma^{\left(eff\right)}=0.25$; red solid line: $\sigma^{\left(eff\right)}=\sigma_{cp}^{\left(eff\right)}\equiv 0.5$; green dash-dotted line: supercritical state, $\sigma^{\left(eff\right)}=0.75.$ Lower panel (**b**): pressure variation $(r^{\left(eff\right)}=r_{P}^{\left(eff\right)}$), $\frac{P^{*}}{P_{0}}=1,\;\frac{∆P_{0}}{P_{0}}=0.1.$ Black solid line: $\sigma^{\left(eff\right)}=0;$ blue dashed line: subcritical state, $\phi_{0}\frac{{\xi_{IN}^{\left(P\right)}}^{2}}{d_{e}^{\left(1\right)}}\frac{a_{NP}}{v_{NP}}$ =0.1; red solid line: $\phi_{0}\frac{{\xi_{IN}^{\left(P\right)}}^{2}}{d_{e}^{\left(1\right)}}\frac{a_{NP}}{v_{NP}}$ =0.27, corresponding to $\sigma^{\left(eff\right)}=\sigma_{cp}^{\left(eff\right)}\equiv 0.5$; green dash-dotted line: supercritical state, $\phi_{0}\frac{{\xi_{IN}^{\left(P\right)}}^{2}}{d_{e}^{\left(1\right)}}\frac{a_{NP}}{v_{NP}}=0.3.$

**Figure 3 nanomaterials-14-00320-f003:**
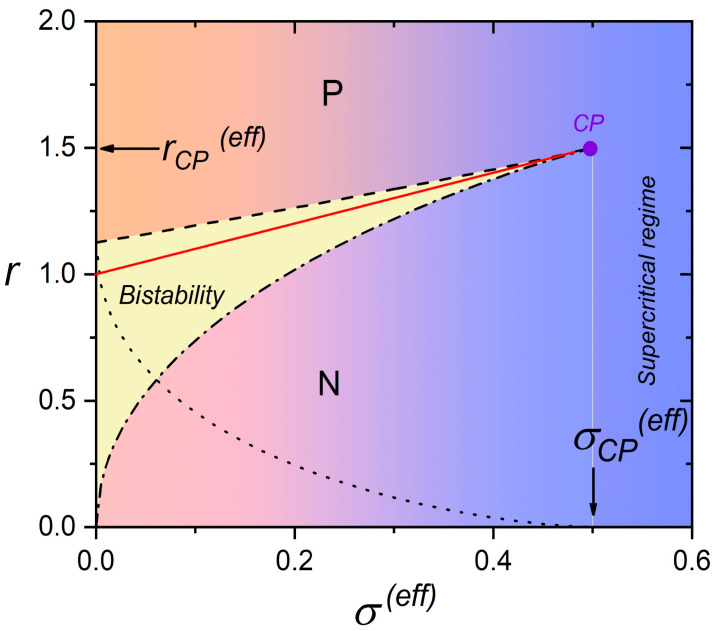
Stability of paranematic (P) and nematic (N) phase upon varying $\sigma^{\left(eff\right)}$. Red solid line: phase transition line; black dashed line: spinodal line $r=r^{\left(eff\right)}=r^{**}(the\;minimum\;corresponding\;to\;the\;nematic\;order\;vanishes)$; black dashed-dotted line: spinodal line $r=r^{\left(eff\right)}{=r}^{*}$ (the minimum corresponding to the paranematic order vanishes); black dotted line: ${r=r}^{**}-r^{*}.$

**Figure 4 nanomaterials-14-00320-f004:**
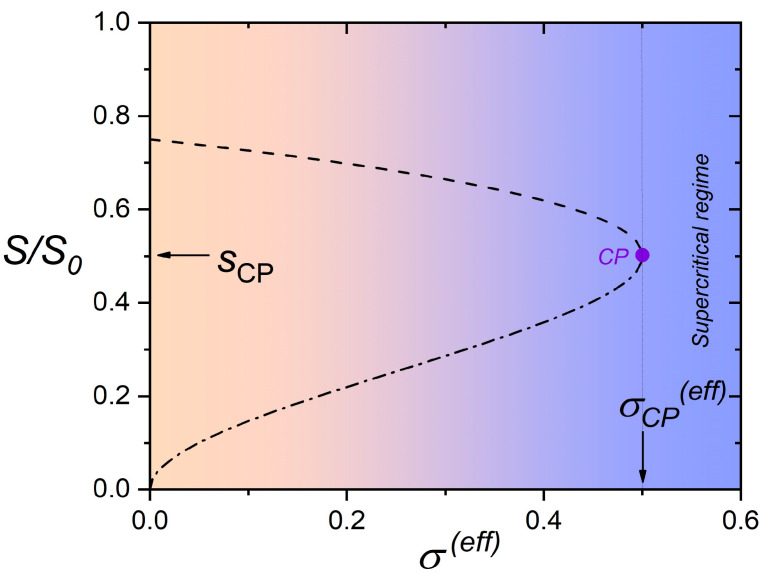
Equilibrium values of the scaled order parameter at spinodal lines. Dash-dotted line: $s^{*}=s(r^{*})$, dashed line: $s^{**}=s(r^{**})$.

## Data Availability

Data are contained within the article.

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
