# Peer review of "Criticality Controlling Mechanisms in Nematic Liquid Crystals"

_nanomaterials, 2024, doi:10.3390/nano14030320_

Round 1
Reviewer 1 Report
Comments and Suggestions for Authors
This is a manuscript that theoretically describes the general mechanism for establishing critical behavior in nematic liquid crystals, and derived a scaled effective free energy expression regarding the impact of different mechanisms on critical behavior. I recommend that this work be published subject to a number of relatively minor issues. The followings are the places required attentions:
1.The format of the references [13] does not meet the requirements of the journal.
2.The introduction of Figures 3 and 4 in the main text is too simplistic, and should be supplemented comprehensively accordingly. In addition, the meanings corresponding to the abbreviations P and N in Figure 3 should be indicated in the title of Figure 3.
3The expressions of the formula for Effective free energy in Section 3 are too cumbersome. It is recommended to simplify them for the convenience of readers to read and understand.
4.If possible, it is recommended to provide a schematic diagram about the impact of nanoparticles on the temperature and pressure induced critical behavior during the first-order I-N transition.
Author Response
We thank the referee for his constructive comments.
Referee comment 1:
The format of the references [13] does not meet the requirements of the journal.
Answer 1:
We corrected the reference [13] format.
Referee comment 2:
The introduction of Figures 3 and 4 in the main text is too simplistic, and should be supplemented comprehensively accordingly. In addition, the meanings corresponding to the abbreviations P and N in Figure 3 should be indicated in the title of Figure 3.
Answer 2:
We improved the text explaining the content of Fig. 3 and 4, see lines 301-308 and lines 350-353.
Referee comment 3:
The expressions of the formula for Effective free energy in Section 3 are too cumbersome. It is recommended to simplify them for the convenience of readers to read and understand.
Answer 3:
We expressed the equations in a simpler way for the convenience of readers, see Eqs.(18) and Eq. (19) (lines 204-207 ), and Eqs.(20) and Eq. (21) (lines 216-220).
Referee comment 4:
If possible, it is recommended to provide a schematic diagram about the impact of nanoparticles on the temperature and pressure induced critical behavior during the first-order I-N transition.
Answer 4:
In Fig.4 the phase transition (red colored) line is shown for a general case. Furthermore, from Eq.(26) and Eq.(27) (lines 252-253) it is evident that the phase transition shifts are roughly linearly dependent on concentration of NPs.
Reviewer 2 Report
Comments and Suggestions for Authors
Dr. Kralj and co-authors theoretically study the generic mechanisms that could establish a critical behavior in nematic liquid crystals. They focus on the impact of nanoparticles (NPs) in homogeneous NP- NLC mixtures and illustrate that in the case of pressure-driven phase changes, lower concentrations are needed to impose critical point conditions in comparison with pure temperature variations. Just some suggestions for this work.
1) In the Introduction, the authors mentioned the universal features and to effectively manipulate and master the CP behavior. It is better to add a scheme for analysis the focus in this work.
2) In the conclusion section, whether it is common to cite the references to support your findings and opinions.
Author Response
We thank the referee for his constructive comments.
Referee comment 1:
In the Introduction, the authors mentioned the universal features and to effectively manipulate and master the CP behavior. It is better to add a scheme for analysis the focus in this work.
Answer:
As suggested by the referee we added a description of the paper in the introduction section, see additional paragraph: lines 72-76.
Referee comment 2:
In the conclusion section, whether it is common to cite the references to support your findings and opinions.
Answer:
As suggested we cite now the relevant references in the conclusion section. We also added a longer discussion on the validity of our derivation, see lines 381-411.
Reviewer 3 Report
Comments and Suggestions for Authors
In the present paper the authors discuss with a standard Landau free-energy approach the effect of nanoparticles on the isotropic to nematic transition. The calculations presented are straightforward and appear to be physically sound.
However, the authors make the very simplistic assumption that the direction of the the electric field is the same as the direction of the induced nematic director n^(j), and so all directionality and the competition between these two directions is lost. It would make the paper much nicer and much more useful if the authors add this competition to their paper and discuss how this affects the phase diagram on the system and ultimately also the directionality of the nematic order parameter Q, as a function of the angle between the electric field and n^(j). I would prefer to recommend only publication of this paper with this additional analysis.
Comments on the Quality of English Language-
Author Response
Referee comment:
However, the authors make the very simplistic assumption that the direction of the the electric field is the same as the direction of the induced nematic director n^(j), and so all directionality and the competition between these two directions is lost. It would make the paper much nicer and much more useful if the authors P 0 WORDS Y   add this competition to their paper and discuss how this affects the phase diagram on the system and ultimately also the directionality of the nematic order parameter Q, as a function of the angle between the electric field and n^(j). I would prefer to recommend only publication of this paper with this additional analysis
Answer:
We thank the referee for his comment. In the revised manuscript we now consider also cases where the nematic director field and electric field are not parallel. Consequently, we generalized equations for general mutual alignment of and : see Eq.(8) and text below it (lines 155-156), Eq.(19) (line 207), Eq.(21) (line 220), Eq.(26) (line 252), Eq.(27) (line 253), and the text below Fig.1 (lines 271-278).